# Interaction between Human Papillomavirus-Encoded E6 Protein and AurB Induces Cell Immortalization and Proliferation—A Potential Target of Intervention

**DOI:** 10.3390/cancers15092465

**Published:** 2023-04-25

**Authors:** Siaw Shi Boon, Yin Ching Lee, Ka Lai Yip, Ho Yin Luk, Chuanyun Xiao, Man Kin Yim, Zigui Chen, Paul Kay Sheung Chan

**Affiliations:** Department of Microbiology, Faculty of Medicine, The Chinese University of Hong Kong, Shatin, New Territories, Hong Kong SAR, China; boonss@cuhk.edu.hk (S.S.B.);

**Keywords:** HPV, E6, Aurora kinase B, carcinogenesis, hTERT, telomerase, AZD1152

## Abstract

**Simple Summary:**

This study identified that Aurora kinase B (AurB), a cellular protein that is upregulated in human cancers, is *a bona fide* interacting partner of HPVE6. HPVE6 complexes with AurB at the C-terminus end of E6, upstream of the E6-PBM. The AurB-E6 complex forms predominantly in the nucleus or mitotic cells. The positive correlation between E6 and AurB protein in HPV-positive cancer cells confers an increased cell proliferation and growth, and the eventual tumour formation. This study also underlined that the AurB-E6 complex could be a therapeutic target. However, the commercially available Aurora kinase inhibitors to date might not be selective toward HPV-positive cancer cells. Hence, this lack should be addressed.

**Abstract:**

The human papillomavirus E6 and E7 oncoproteins interact with a different subset of host proteins, leading to dysregulation of the apoptotic, cell cycle, and signaling pathways. In this study, we identified, for the first time, that Aurora kinase B (AurB) is a *bona fide* interacting partner of E6. We systematically characterized the AurB-E6 complex formation and its consequences in carcinogenesis using a series of *in vitro* and cell-based assays. We also assessed the efficacy of Aurora kinase inhibitors in halting HPV-mediated carcinogenesis using *in vitro* and *in vivo* models. We showed that AurB activity was elevated in HPV-positive cells, and this correlated positively with the E6 protein level. E6 interacted directly with AurB in the nucleus or mitotic cells. A previously unidentified region of E6, located upstream of C-terminal E6-PBM, was important for AurB-E6 complex formation. AurB-E6 complex led to reduced AurB kinase activity. However, the AurB-E6 complex increased the hTERT protein level and its telomerase activity. On the other hand, AurB inhibition led to the inhibition of telomerase activity, cell proliferation, and tumor formation, even though this may occur in an HPV-independent manner. In summary, this study dissected the molecular mechanism of how E6 recruits AurB to induce cell immortalization and proliferation, leading to the eventual cancer development. Our findings revealed that the treatment of AZD1152 exerted a non-specific anti-tumor effect. Hence, a continuous effort to seek a specific and selective inhibitor that can halt HPV-mediated carcinogenesis should be warranted.

## 1. Introduction

Human papillomavirus (HPV) contributes to approximately 5% of human cancers and accounts for 7.5% of cancer deaths in women [1]. Among the over 400 HPV genotypes identified officially, only 13 are classified as “high-risk” types, owing to their ability to cause malignancy. HPV16 and 18 are the most prevalent types contributing to over 70% of cervical cancers [2], as well as other anogenital and head-and-neck cancers. While the incidence of HPV-related cervical cancer is declining, there is an alarming trend of an increase in the incidence of HPV-associated head-and-neck cancers, particularly oropharyngeal cancer [3,4].

Upon HPV infection, where the viral particles gain access to the primitive basal layer of keratinocytes, the viral genome is expressed and maintained alongside host cell differentiation [5]. In low-grade lesions, full viral genome expression can be detected. As the lesion progresses, the HPV genome can integrate into that of the host, leading to the steady overexpression of the E6 and E7 oncoproteins. This marks the eventual carcinoma [6]. During this process, E6 and E7 perturb multiple host cell processes: E6 degrades p53 [7,8], Bak [9], and PSD95/Dlg/Zona-occludens-1 (PDZ) proteins [10,11], leading to dysregulation of apoptosis, cell cycle, and cell polarity. E7 concurrently degrades pRB and its related pocket proteins [12,13], resulting in the perturbation of E2F-directed transcriptional transactivation [14,15] and the shortening of the G1/S transition [16]. E6 and E7 activities combined can activate the DNA damage response [17], induce a global increase in the expression of cyclin-dependent protein kinases (CDKs) [16], and increase genome instability [18,19].

Recent reports showed that members of the Aurora kinase family of serine/threonine kinases are involved in HPV-mediated carcinogenesis. In mammalian cells, there are three Aurora kinases: Aurora kinase A (AurA), B (AurB), and C (AurC) [20]. Aurora kinases are often upregulated in human cancers, including cervical and head-and-neck cancers (HNC) [21,22]. Upregulation of Aurora kinases often causes dysregulation of the G2/M checkpoint [23], enhances the RAS/MEK/ERK/MAPK signaling cascade [24], and increases telomerase activity [25,26]. 

Previous and recent studies showed that E6 encoded by HPV16 binds to AurA [27,28], and this may lead to increased Aurora kinase A levels in head-and-neck cancers [22]. However, the other Aurora kinases have not been investigated. In view of this, this study focused on the association between E6 encoded by high-risk HPV types, including HPV16 and HPV18, and AurB. We identified AurB as a *bona fide* partner of E6. We performed a series of *in vitro* and cell-based assays to investigate the physical association between E6 with AurB. We then delineated the consequences of AurB-E6 complex formation on key oncogenic pathways, including activation of human telomerase reverse transcriptase (hTERT) protein level and its telomerase activity. We also assessed and compared the efficacy of Aurora kinase inhibitors in disrupting hallmarks of cancer phenotype using 3-dimensional (3D) *in vitro* and *in vivo* models.

## 2. Materials and Methods

### 2.1. Plasmids

HPV-E6 expression constructs, namely pGW1:HA-HPV-16 and -18 E6, pGEX2T:HPV-16 and -18 E6 (encoding GST-HPV16 and HPV18 E6 conjugated proteins, respectively), were generously gifted by Dr. Lawrence Banks, ICGEB. These expression constructs were used in *in vitro* binding, co-immunoprecipitation, and transient expression assays. pcDNA:Flag-Aurora kinase B (FL-AurB) was a generous gift from Prof. Li-Yuan Yu-Lee, Baylor College. For subcellular localization and immunoprecipitation assays, pGW1:HA-HPVE6s and pcDNA:Flag-AurB mammalian expression constructs were used. 

Using online Scansite freeware [29], we predicted 4 potential AurB recognition sites within 16 E6: S89, T140, S145, and S150. These amino acids were mutated to Alanine (A), as depicted schematically in Appendix A, using the GeneArt^TM^ site-directed mutagenesis system (Invitrogen, Carlsbad, CA, USA) according to the manufacturer’s protocol. Primer pairs used to generate these expression constructs are listed in Appendix A. The resulting plasmids were all verified by DNA sequencing and transformed into *E. coli* strain DH5-α.

### 2.2. Cell Lines

HeLa (HPV18 cervical cancer cells), CaSki (HPV16 cervical cancer cells), Human embryonic kidney (HEK) 293 (HPV-null epithelial cells), C33A (HPV-null cervical cancer cells) and U-2 OS (HPV-null human osteosarcoma cells) were purchased from the American Type Culture Collection (ATCC) and maintained in Dulbecco’s modified Eagle medium (DMEM) supplemented with 10% fetal bovine serum (FBS, GIBCO) at 37 °C in a humidified incubator with 5% CO_2_. Human primary normal epidermal keratinocytes, adult (HeKa) cell line was purchased from ThermoFisher Scientific (Waltham, MA, USA) and maintained in EpiLife^TM^ medium supplemented with human keratinocyte growth supplement (HKGS). Identities of these human cell lines were validated by short tandem repeat (STR) profiling using the AmpFlSTR Identifiler^®^ Plus PCR Amplification Kit (ThermoFisher Scientific) with the Applied Biosystems 3500 Series Genetic Analyzer and analyzed by GeneMapper^®^ Software 5 (Applied Biosystems, Bedford, MA, USA). The STR profile of all cell lines showed >88% concordance with their reference profiles in the ATCC cell line database.

### 2.3. Immunoprecipitation Assay

HEK293 cells (7 × 10^5^) were seeded onto 10-cm dishes and transfected with 10 μg of hemagglutinin (HA)-tagged HPV18 or HPV16 E6. At 24 h post-transfection, cells were treated with 200 ng/mL nocodazole (Sigma, St. Louis, MO, USA) for 18 h. Cells were collected and lysed by using E1A buffer [50 mM NaCl, 0.1% NP-40, and 50 mM HEPES (pH 7.0)], with gentle syringing, and then placed on ice for 20 min. The cell lysate was then centrifuged at 14,000 rpm for 10 min, and the supernatant was incubated with 20 μL of monoclonal Pierce^TM^ anti-HA agarose beads (ThermoFisher Scientific) at 4 °C for 3 h. Samples were washed thrice with E1A buffer and then subjected to western blot analysis.

### 2.4. Downregulation of HPV Oncoproteins Using siRNA

Downregulation of HPV18 E6 and E6/E7 was performed as described previously [30]. In brief, 2 × 10^5^ HeLa cells were seeded into 6-well plates. The cells were transfected with short interference RNA (siRNA) using Lipofectamine LTX (Invitrogen) for 72 h. The total cell lysate was collected and subjected to western blotting.

### 2.5. Fusion Protein Purification and In Vitro Binding Assays

Transformants harboring pGEXT2T:HPVE6s constructs were grown overnight. The culture was then inoculated into a fresh medium and incubated for 1 h. Recombinant protein expression was then induced by the addition of isopropyl-β-D-thiogalactopyranoside (IPTG, Sigma) to a final concentration of 1 mM and incubated for a further 3 h. The bacteria were harvested and lysed with cold 1× PBS containing 1% Triton X-100 and sonicated twice for 30 s (sec). The supernatants were collected and incubated with glutathione-conjugated agarose resin on a rotating wheel overnight at 4 °C. After extensive washing, the amount of immobilized GST fusion proteins were analyzed by sodium dodecyl sulfate-polyacrylamide gel electrophoresis (SDS-PAGE) and stained by GelCode^TM^ Blue Stain Reagent (ThermoFisher Scientific). 

AurB protein was translated *in vitro* using the TNT^®^ Coupled Reticulocyte Lysate System (Promega, Madison, WI, USA), according to the manufacturer’s recommendation. Equal amounts of *in vitro*-translated AurB were added to purified GST fusion proteins and incubated for 1 h at room temperature. After being washed thrice with PBS containing 0.1% Tween 20, the bound proteins were subjected to SDS-PAGE and western blotting for the detection of AurB.

### 2.6. Immunofluorescence Assay

This assay is carried out as described previously [30]. Approximately 2 × 10^5^ U-2 OS cells were plated onto coverslips. The cells were either mock-transfected or transfected with HA-tagged 16 E6 or E7. Cells were fixed with ice-cold absolute methanol 24 h after transfection and incubated with primary antibody against HA (Roche) and Aurora B (Cell Signaling), followed by Alexa Fluor^®^ 568-conjugated anti-rabbit secondary antibody and Alexa Fluor^®^488-conjugated anti-mouse secondary antibodies (ThermoFisher Scientific), and counterstained with 4′,6-diamidino-2-phenylindole (DAPI). The subcellular location of HPV16 E6 or E7 and AurB were examined under a fluorescence microscope (Leica, Wetzlar, Germany).

### 2.7. ADP-Glo^TM^ AurB Kinase Assay

HEK293 cells were transfected with FL-AurB. The cell lysate was collected, and the full-length human AurB was then purified from the cells. The kinase reaction was carried out by incubating 10 μL of purified AurB (approximately 2 μg) with equal amounts of purified HPVE6s GST fusion proteins in the presence of 30 μL of kinase reaction buffer containing 25 mM Tris (pH 7.5), 10 mM MgCl_2_, 3.5 mM NaCl and 75 μM ATP for 1 h at 25 °C. The AurB kinase activity was measured using ADP-Glo™ Kinase Assay (Promega), according to the manufacturer’s recommendations. Briefly, the kinase reaction was mixed with ADP-Glo™ Reagent at a ratio of 1:1 and incubated in a white base 384-well plate for 1 h. Following this, one volume of Kinase Detection reagent was added and incubated for an additional 1 h at 25 °C in the dark. The amount of luminescence ADP produced was measured using VICTOR Multilabel Plate Reader (PerkinElmer, Waltham, MA, USA).

### 2.8. Telomeric Repeat Amplification Protocol (TRAP) Assay

A PCR-based assay using TRAPeze^®^ Telomerase Detection Kit (Merck, Rahway, NJ, USA) was employed to detect telomerase activity according to the manufacturer’s recommendations. In brief, 5000 U-2 OS (HPV-null osteosarcoma cell lines lack hTERT and telomerase activity) cells were either mock-transfected or with either HA-tagged hTERT, HA-tagged HPV16 E6, Flag-tagged AurB alone, or co-transfected. The cells were collected and lysed. Cell lysate was then subjected to a one-step PCR amplification using master mix provided by the kit. The products were resolved on a nondenaturing polyacrylamide gel. The gel was stained using SYBR Safe DNA Gel Stain (Invitrogen, Carlsbad, CA, USA) and viewed using the ChemiDoc^TM^ Imaging Systems (Bio-Rad, Hercules, CA, USA).

### 2.9. Quantitative Polymerase Chain Reaction (qPCR)

The cells were collected, and total RNA was extracted using RNeasy Mini Kit (QIAGEN) according to the protocol recommended by the manufacturer. The cDNA was synthesized by adding a total of 1 µg RNA for reverse transcription using S SuperScript^TM^ III Reverse Transcriptase (Invitrogen, Carlsbad, CA, USA). Quantitative PCR (qPCR) will be performed in duplicate by adding 10 µM primers and Luna^®^ Universal qPCR Mastermix (NEB, Ipswich, MA, USA) in the real-time PCR system (Applied Biosystem, Bedford, MA, USA). The primers for AurB, E6, and GAPDH (housekeeping control) are listed below (Table 1): 

### 2.10. Western Blotting

Total protein was collected by lysing cell pellets with 2× sodium dodecyl sulfate (SDS) sample buffer and boiled at 95 °C for 5 min. Total protein lysate was resolved by SDS-PAGE and transferred onto a polyvinylidene fluoride (PVDF) membrane (GE Healthcare). The membrane was then probed with specific primary antibodies overnight at 4 °C: phospho-Aurora kinase A (T288)/B (T232)/C (T198), Aurora kinase B (Cell Signaling Technology), HPV16E7 (Cervimax), p53 (DO-1), HA (Roche), Flag^®^ M2 (Sigma), hTERT (Abcam), Ras, MEK1/2, ERK1/2, GAPDH and β-actin (Santa Cruz Biotechnology, Santa Cruz, CA, USA). Subsequently, the immunoblots were incubated with the appropriate HRP-conjugated secondary antibodies. Blots were visualized using Clarity^TM^ Western ECL Substrate (Bio-Rad), and images were captured using a ChemiDoc™ Imaging System (Bio-Rad, Hercules, CA, USA). β-actin or glyceraldehyde 3-phosphate dehydrogenase (GAPDH) served as loading controls for the blots. Band intensities were quantified by Image Lab™ Software (Bio-Rad, Hercules, CA, USA) and normalized using the corresponding loading controls.

### 2.11. Cell Proliferation Assay

The cell proliferation was determined by WST-8 [2-(2-methoxy-4-nitrophenyl)-3-(4-nitrophenyl)-5-(2,4-disulfophenyl)-2H-tetrazolium] assay using a Cell Counting kit-8 (CCK-8; Dojindo, Shanghai, China). In brief, C33A, HeLa, and CaSki cells were seeded into 96-well plates at a density of 2 × 10^3^ cells/well in 100 μL of complete medium. The cells were either treated with vehicle or AZD1152 (0, 12.5, 25, 50, 100, 200, 400, and 800 nM) for 24 h. WST-8 (10 μL) was added to each well and incubated for 1 h at 37 °C. The optical density (OD) was measured at 450 nm using VICTOR 3 Multilabel Plate Reader (PerkinElmer). 

### 2.12. Organotypic Raft Culture

A day prior to the seeding of C33A, CaSki, and HeLa cells, dermal equivalent (DE) was prepared in transwell chambers. The DE was made up of a collagen mix containing NIH3T3 fibroblasts (2 × 10^5^ cells) suspended in 9X DMEM, 10% FBS, 1× reconstitution buffer (26 mM sodium bicarbonate, 200 mM Hepes), and 4 mg/mL Rat tail collagen I (Enzo Life Sciences, Farmingdale, NY, USA). C33A, CaSki, and HeLa cells (3 × 10^5^ cells) were suspended in DMEM/F-12 media (GIBCO) containing 10% FBS, 5 µg/mL insulin, 8.4 ng/mL cholera toxin, 2.4 µg/mL adenine, 0.4 µg/mL hydrocortisone, 10 ng/mL epidermal growth factor and 100 µg/mL bovine serum albumin (BSA). The culture media was replenished every other day for 14 days. For treatment, DMSO or 300 nM AZD1152 was added into the culture media for an additional 7 days. The organotypic culture was then collected, fixed in 4% paraformaldehyde, and processed for H&E staining. 

### 2.13. In Vivo Xenograft Model

Animal ethics and licenses were obtained prior to conducting this experiment. Immortalized primary baby rat kidney epithelial cells (BRK) and HeLa cells (approximately 5 × 10^6^ cells) were resuspended with 200 μL of DMEM and injected subcutaneously over the flank of 4- to 5-week-old female athymic nude mice (12 mice for DMSO treated group; 18 for AZD1152 treated group) supplied by LASEC, the Chinese University of Hong Kong. Mice body weight and tumor size were monitored and recorded every other day for 2 weeks. Upon reaching palpable tumors, the littles were given 50 mg/kg of AZD1152 dissolved in sodium carboxymethyl cellulose (Viscosity: 800–1200 mPa.s) via the oral route. In the vehicle control group, the littles were injected with DMSO. Carcasses were disposed of by LASEC. Tumor volume was estimated using the formula (a × b^2^) × 0.5236, where “a” refers to the largest dimension of the tumor and “b” is the perpendicular diameter [33]. The tumors were paraffin-embedded and sectioned for hematoxylin and eosin (H&E) staining.

### 2.14. Statistical Analysis

Data presented were performed at least in triplicates of independent experiments. Data were expressed as mean ± SEM with statistical analyses being performed using GraphPad Prism 7. The band intensities of immunoblots and tumor burden of HPV-positive and negative-bearing mice were compared with their respective controls using a *t*-test. The viability of cells upon drug treatments was analyzed via non-linear regression using GraphPad Prism. A *p*-value of <0.05 was considered statistically significant.

## 3. Results

### 3.1. AurB Activity Was Increased in HPV-Positive Cells

Aurora kinases (AurA, B, and C) are activated when they are phosphorylated (residues T288 on AurA, T232 on AurB, and T198 on AurC). Knowing this, we compared the phosphorylated form of Aurora kinases in HPV-positive and HPV-negative cells to reflect their association with HPV oncoproteins. We examined total cell lysates from HeKa (HPV-null primary human keratinocytes), C33A (HPV-null human cervical cancer cells), HeLa (HPV18-positive cervical cancer cells), and CaSki (HPV16-positive cervical cancer cells) by western blotting. As shown in Figure 1A, we observed an increase in the level of AurB phosphorylated at T232 [pAurB (T232) or herein refers as pAurB] in HPV-positive cells (3-fold in HeLa; 1.6-fold in CaSki cells) compared with HPV-null cells. The level of phosphorylated AurA was unchanged, while the phospho-AurC levels were reduced in CaSki cells. We also noted a higher level of total AurB protein in cancer cells (C33A, HeLa, and CaSki) than in normal keratinocytes (HeKa). The level of AurB protein in HeLa (1.5-fold) and CaSki (1.8-fold) was higher than that of C33A. The higher level of pAurB indicates that HPV-positive cancer cells had a higher AurB activity than HPV-negative cells, suggesting that HPV may induce an increased level of AurB activity in HPV-positive cells. 

### 3.2. The Level of AurB Protein Correlated Positively with HPVE6

Next, we asked if the increased level of AurB protein has a correlation with either E6- and/or E7-encoded by HPV16 and 18. Intriguingly, we observed that the level of AurB protein and its activity in HPV-positive cancer cells correlated positively with the level of HPVE6 protein through the following two approaches: (i)Overexpression of HPV oncoproteins in HPV-null cells

Firstly, we overexpressed HA-tagged E6-encoded by HPV16 and HPV18 in HEK293 cells. From our immunoblots (Figure 1B), we observed an increased level of pAurB, while the expression level of total AurB was unchanged. 

(ii)Depletion of HPV oncoproteins in HPV-positive cells

Secondly, we depleted the expression of E6 and/or E7 in HeLa cells by transfecting short interference RNA (siRNA) specifically against HPV18 E6 alone (si18E6) or both E6 and E7 (si18E6/E7). As shown in the immunoblots in Figure 1C, downregulation of E6 led to dramatically decreased levels of both pAurB (41.7% ± 10.7%, *p* < 0.05) and total AurB (37.6% ± 6.8%, *p* < 0.01), when compared with cells transfected with siControl (siCtrl). Similarly, when both E6 and E7 were downregulated, pAurB (20.3% ± 4.5%, *p* < 0.01) and total AurB (20.5% ± 5.3%, *p* < 0.01) were also reduced. The downregulation of either E6 (1.4 ± 0.6, *p* = 0.565) or both E6 and E7 (0.9 ± 0.4, *p* = 0.878) did not affect the AurB mRNA expression. This result indicates that the reduced pAurB and AurB occur at the protein level, and this depends upon the level of E6 protein.

### 3.3. AurB Interacted Directly with HPVE6 at the C-Terminus of HPVE6, Independent of E6-PBM

As we observed an association between AurB and HPVE6, we next wanted to investigate if this association occurs through direct interaction between AurB and E6. We addressed this through co-immunoprecipitation and *in vitro* binding assays. From both approaches, we observed that AurB interacted directly with HPVE6. 

(i)AurB co-immunoprecipitated with HPVE6 but not HPVE7.

After transfecting HA-tagged E6- and E7-encoded by HPV16 in HEK293 cells, the cells were treated with nocodazole, which is an anti-mitotic agent that can induce an increase in endogenous AurB protein level (Figure 2A,B). We then immunoprecipitated E6 and E7 from cell extracts using HA-conjugated agarose resin and examined the bound AurB via western blotting. We observed that AurB, upon treatment with nocodazole, bound to HA-HPV16E6 but not to HA-HPV16E7 (Figure 2A). We also detected AurB bound to HA-HPV18E6 in HEK293 cells (Figure 2B). This indicates that AurB interacts with both E6-encoded HPV16 and 18 in HEK293 cells but not with 16E7. 

(ii)E6-encoded by HPV16 and 18 bound directly to AurB.

We next investigated whether the interaction between AurB and E6 is direct. To do this, we performed an *in vitro* binding assay, where *in vitro*-translated AurB was incubated with purified GST-tagged E6 fusion proteins encoded by HPV16 or 18, and the bound AurB was detected via western blotting. As shown in Figure 2C, AurB is bound to purified GST-HPV16 and -HPV18E6 fusion proteins. 

(iii)AurB bound to E6 at the C-terminus of E6, independent of E6-PBM.

We then extended our analysis to examine the key amino acids of E6 that interact with AurB. Through Scansite, it was predicted that amino acid residues serine (S) 89, threonine (T) 140, S145, and S150 of HPV16E6 (herein refers as E6) harbored AurB recognition motif (Appendix A). Therefore, these amino acid residues of E6 were mutated to Alanine (A). After performing similar *in vitro* binding assays described above, we observed that all the E6 mutants showed a reduced binding affinity to AurB (Figure 2D(i,ii)). Among these mutants, E6 S145A bound the weakest to AurB (27% ± 7%; *p* < 0.001), followed by S89A (44% ± 14%; *p* < 0.05), T140A (45% ± 11%; *p* < 0.01) and S150A (57% ± 7%; *p* < 0.01). 

As the amino acids S89 to S150 of E6 are located outside, but in very close proximity to the extreme C-terminus end of the PDZ binding motif of E6 (E6-PBM), we wanted to investigate if other amino acids and the integrity of E6-PBM might be important for AurB-E6 complex formation, we, therefore, included E6 mutant in which PBM is depleted (E6ΔPBM) and performed a similar *in vitro* binding assay. Intriguingly, we found that the E6ΔPBM mutant bound to *in vitro*-translated AurB was, in fact, 3-fold stronger than the E6 wild-type (Figure 2E(i,ii)).

These results indicate that AurB interacts directly with HPV16 and HPV18 E6 in an E6-PBM-independent manner. The amino acid residues at the C-terminus of HPV16E6 (140–150), a region upstream of E6-PBM, are the most critical for the AurB-E6 complex formation. 

### 3.4. HPVE6 Formed a Complex with AurB in the Nucleus

Subsequently, we examined the subcellular co-localization of AurB and E6 using an immunofluorescence assay. We ectopically expressed HA-tagged HPV16 E6 or E7 in U-2 OS, an HPV-null human osteosarcoma cell line. This cell line was chosen due to the following reason: (i) the cell is easily transfected with detectable protein level; (ii) morphologically, the cells are flat and have a distinctive nuclear and cytoplasmic region; (iii) the cells lack the expression of hTERT, which is a target of AurB and E6. We found that AurB co-localized with HPV16E6 in the nucleus of these cells. It appeared that E6 co-localized with AurB in the mitotic cells (Appendix A). We observe a fairly weak or none overlapped expression of E7 with AurB. This result indicates that AurB interacts with E6 directly, and the AurB-E6 complex is formed predominantly in the nucleus and mitotic cells. 

### 3.5. E6 Inhibited AurB Kinase Activity

Upon gaining knowledge that AurB expression correlates positively with the expression of HPVE6 through direct interaction between AurB and E6, we were intrigued to know if such an association affects the functions of AurB. As one of the obvious functions of AurB is to act as a mitotic kinase, we performed a sensitive luminescence-based ADP-Glo^TM^ kinase assay. This assay examines the level of consumed ATP (ADP) following kinase activity, with a higher amount of ADP indicating higher kinase activity and vice versa. We incubated purified AurB with purified GST-16E6 fusion proteins. As AurB can undergo autophosphorylation, purified AurB was incubated with purified GST protein alone to serve as a positive control. We found that, in the presence of GST-16E6 fusion proteins, the AurB activity was reduced by more than half (Figure 2F). Among the E6 mutants, E6 T140A (24.4% ± 4.0%; *p* < 0.0001) and E6 S145A (38.0% ± 5.8%; *p* < 0.001) reduced AurB activity more dramatically, followed by wild-type E6 (44.0% ± 9.6%; *p* < 0.01), E6 S150A (46.9% ± 9.8%; *p* < 0.01) and E6 S89A (48.7% ± 16.1%; *p* < 0.05). This indicates that the binding of E6 to AurB, disregarding the affinity of binding of E6 to AurB, can reduce the AurB kinase activity.

### 3.6. E6 Did Not Perturb the Function of AurB in Regulating hTERT Protein

As both AurB and E6 independently perturb hTERT, we then investigated if the AurB-E6 association could affect hTERT protein level and activity. In order to do this, an HPV-null U-2 OS (an osteosarcoma cell line that lacks hTERT expression) was used. HA-tagged hTERT was transfected into U-2 OS cells either alone or together with Flag-tagged AurB (FL-AurB) and/or HA-tagged 16E6 (HA-16E6). Cell lysates were collected, and the protein expression level of hTERT was ascertained via western blotting. As shown in Figure 3A, co-expression of hTERT with AurB and E6 (460.5% ± 109.7%, *p* < 0.05) or with AurB (412.4% ± 161.0%, *p* < 0.05) led to an increased hTERT expression by at least 4-folds when compared with the cells with hTERT expressed alone. This result indicates that E6 does not perturb the function of AurB in mediating an increase in hTERT expression. 

### 3.7. The Association of AurB and E6 Led to Increased Telomerase Activity

Next, we assessed if the telomerase activity in hTERT-expressing cells was perturbed by AurB-E6 by performing a PCR-based TRAPeze telomerase detection assay. U-2 OS cells were transfected with FL-AurB and HA-16E6, or with HA-hTERT and FL-AurB, or HA-hTERT, FL-AurB, and HA-16E6. As shown in Figure 3A(iii) and S3A, at least a 2-fold increment of telomerase activity was detected in cells with AurB and E6 (265.4% ± 45.8%, *p* < 0.05), hTERT, and AurB (282.0% ± 65.9%, *p* < 0.05), or hTERT was co-expressed with AurB and E6 (251.3% ± 30.4%, *p* < 0.01) when compared with the basal telomerase activity in the mock-transfected U-2 OS cells. This result demonstrates that E6 does not perturb the function of AurB in regulating telomerase activity. More strikingly, we also observed that AurB and E6 overexpression were able to increase the basal level of telomerase activity.

Next, we were intrigued to understand the interplay between hTERT and Ras/MEK/ERK pathway in the presence of AurB and E6. This is because hTERT can inhibit the Ras/MEK/ERK pathway, which is one of the key oncogenic pathways [34]. We found that hTERT, regardless of when co-expressing with AurB or E6, can reduce the levels of Ras/MEK/ERK pathway (Appendix A). When hTERT co-expressed with AurB and E6, the levels of Ras, MEK1/2, and ERK1/2 were similar to that of in the mock-transfected cells, indicating that the AurB-E6 complex may override hTERT inhibition on Ras/MEK/ERK axis, thereby allowing cell survival.

Altogether, these results indicate that the AurB-E6 association can maintain a high hTERT protein level. More importantly, AurB and E6 can accelerate telomerase activity synergistically in HPV-positive cancer cells, which is critical for E6 to induce immortalization during cancer progression. 

### 3.8. Inhibition of Aurora Kinase B Reduced Telomerase Activity

As we observed a positive correlation between AurB and E6 that could promote oncogenic events at the molecular level and telomerase activity, we next asked if AurB inhibition could influence AurB-mediated telomerase activity of hTERT. Prior to assessing this, we examined the effect of an AurB-specific inhibitor (AZD1152) and pan-Aurora kinases inhibitor (Reversine, AMG900, and Tozasertib) on the expression of (a) phosphor- and total AurB; (b) HPV16E7; (c) p53 (as a surrogate marker for E6 expression in HPV-positive cells); and (d) apoptotic pathway activation. We treated C33A and CaSki cells with low (the minimal concentrations sufficient to induce changes in cellular proliferation) and high concentrations (IC_50_ value of the inhibitors) of the inhibitors: 30 nM or 300 nM of AZD1152, 5 μM or 30 μM of Reversine, 5 nM or 25 nM of AMG900, and 1.8 nM or 18 nM of Tozasertib.

Phosphor- and total AurB proteins

Firstly, we delineated the effect of these inhibitors on phosphor-AurB and total AurB in C33A and CaSki. As shown in Figure 3B(i,ii), treatment of CaSki with 30 nM (0.24 ± 0.09, *p* < 0.01) and 300 nM (0.50 ± 0.12, *p* < 0.05) of AZD1152, 5 μM of Reversine (0.35 ± 0.11, *p* < 0.01), 5 nM (0.25 ± 0.09, *p* < 0.01) and 25 nM (0.34 ± 0.28, *p* < 0.01) AMG900, and 1.8 nM of Tozasertib (0.34 ± 0.19, *p* < 0.05) reduced phosphor-AurB protein level. While treatment of C33A with 300 nM of AZD1152 (0.37 ± 0.17, *p* < 0.05), 25 nM of AMG900 (0.28 ± 0.15, *p* < 0.01), 1.8 nM (0.54 ± 0.12, *p* < 0.05) and 18 nM (0.31 ± 0.03, *p* < 0.0001) of Tozasertib also reduced phosphor-AurB protein level. However, 30 nM of AZD1152, 5 μM and 30 μM of Reversine, and 5 nM of AMG900 did not affect phosphor-AurB protein level. 

Intriguingly, we observed a different pattern of total AurB protein levels in C33A and CaSki upon AurB inhibition. As shown in Figure 3B(i,iii), treatment of CaSki with 300 nM of AZD1152 (0.55 ± 0.17, *p* < 0.05) reduced total AurB protein level, while treatment of 25 nM of AMG900 (1.07 ± 0.11, *p* < 0.01) induced an increased AurB protein level in CaSki cells. When C33A was treated with 300 nM of AZD1152 (2.32 ± 0.53, *p* < 0.05) and 25 nM of AMG900 (2.63 ± 0.55, *p* < 0.001), the level of AurB protein was increased. 

b.HPV16E7 protein

Secondly, we were intrigued to find that treatment of CaSki cells with all the inhibitors at a higher concentration led to a reduced HPV16E7 protein expression: 300 nM of AZD1152, 0.07 ± 0.25, *p* < 0.0001; 30 μM of Reversine, 0.20 ± 0.03, *p* < 0.0001; 25 nM of AMG900, 0.10 ± 0.04, *p* < 0.0001; and 18 nM of Tozasertib, 0.13 ± 0.04, *p* < 0.0001) (Figure 3B(i,iv)). While low concentrations treatment of CaSki with 5 μM of Reversine (0.32 ± 0.11, *p* < 0.001) and 5 nM of AMG900 (0.57 ± 0.09, *p* < 0.01) were also sufficient to significantly reduce the level of HPV16E7 protein. 

c.p53 protein

Thirdly, when we assessed the level of p53 protein (Figure 3B(i,v)), we observed an increased p53 protein level when CaSki cells were treated with 300 nM of AZD1152 (2.56 ± 0.72, *p* < 0.05), 5 μM (2.50 ± 0.64, *p* < 0.05) and 30 μM (4.02 ± 0.88, *p* < 0.01) of Reversine, 25 nM of AMG900 (5.64 ± 1.49, *p* < 0.05), and 18 nM of Tozasertib (5.48 ± 1.51, *p* < 0.01). While treatment of C33A cells with 25 nM of AMG900 (1.44 ± 0.13, *p* < 0.05) treatments induced an increased level of p53 protein. Treatment with AZD1152, Reversine, and Tozasertib, either at low or high concentrations, did not affect the level of p53 proteins in C33A cells. Similarly, C33A cells treated with 5 nM of AMG900 also did not affect the level of p53 protein.

d.Apoptosis activation

From the above results, we observed that 300 nM of AZD1152 treatment reduced the protein levels of phosphor- and total AurB and HPV16E7, as well as increased the level of p53 protein), and knowing that AZD1152 is an anti-tumor agent, we next wanted to study if 300 nM of AZD1152 could activate the apoptotic pathway. We treated HPV-positive (HeLa and Caski) and -negative (C33A) cancer cells with 300 nM of AZD1152 for 3 days. We collected total cell lysate on each day of the treatment and analyzed the expression levels of caspase-8 and p53 via western blotting. We observed an increase in caspase-8 and p53 proteins in both HeLa and CaSki cells when these cells were treated with 300 nM of AZD1152 (Appendix A). Interestingly, we also observed increased levels of these proteins in a treatment day-dependent manner, in which the lowest expression of caspase-8 and p53 was observed on Day 0 and the highest expression on Day 3. This increment was not obvious in C33A cells treated with 300 nM of AZD1152. On the contrary, treatment of C33A with 300 nM of AZD1152 reduced the expression of caspase-8 from Day 0 to Day 3. These results indicate that HPV-positive cells are more susceptible to cell death induced by treatment with 300 nM AZD1152 compared with HPV-null cells. 

We then wanted to scrutinize the effect of AZD1152 treatment on telomerase activity using similar experimental settings as described in Figure 3A. We also included HeLa cells, which are known to carry a high endogenous telomerase activity, mock-transfected with a pcDNA3.1 vector. To further demonstrate the inhibitory effect of Aurora kinase inhibitors on telomerase activity, HA-hTERT was overexpressed in HeLa cells. After 24 h, the cells were treated with 300 nM of AZD1152.

Under the treatment of 300 nM of AZD1152, we observed a higher level of telomerase activity in cells with hTERT, AurB, and E6 co-expressed (120.4% ± 5.2%, *p* < 0.05) when compared with the basal telomerase activity in the mock-transfected U-2 OS cells (Figure 3C and Appendix A). Whereas a similar telomerase activity was detected in cells with AurB and E6 (118.9% ± 16.8%, *p* = 0.32) or AurB and hTERT (109.2% ± 9.6%, *p* = 0.39) expressed when compared with the mock-transfected U-2 OS cells. Even though a higher telomerase activity was detected in mock-transfected HeLa cells (155.5% ± 16.8%, *p* < 0.05) and hTERT-transfected HeLa cells (208.5% ± 35.3%, *p* < 0.05) compared to mock-transfected U-2 OS cells, there was no significant difference in the telomerase activity between mock- or hTERT-transfected HeLa cells (*p* = 0.25). These results show that AZD1152 treatment inhibits telomerase activity when AurB and E6 are co-expressed in U-2 OS cells. However, the treatment is insufficient to suppress telomerase activity orchestrated by hTERT-AurB-E6 in HPV-positive cancer cells.

### 3.9. Treatment with AZD1152 Suppressed the Cell Proliferation and Tumour Formation in an HPV-Independent Manner

We performed a cell counting kit-8 (CCK8) assay to assess the ability of viable cells to digest tetrazolium salt into orange formazan after the cells were treated with AZD1152. We found that 206 nM (Appendix A(i)), 151 nM (Appendix A(ii)), and 179 nM (Appendix A(iii)) of AZD1152 achieved 50% inhibition of the viability of HeLa, CaSki, and C33A cells, respectively. These results show that Aurora kinase B inhibition, following treatment with 300 nM of AZD1152, can inhibit the proliferation of both HPV-null and -positive cells. The inhibitory effect exerted by AZD1152 is not specific toward HPV-positive cancer cells. 

We then wanted to recapitulate the efficacy of AZD1152 treatment in inhibiting tumor formation using biologically relevant 3-dimensional organotypic culture and animal models. Therefore, we addressed this by performing the following assays:

(i)Three-dimensional organotypic raft culture

We grew C33A, HeLa, and Caski cells in an organotypic raft culture, where these cells were seeded onto an epidermal equivalent with a collagen layer containing fibroblast cells and supplemented with growth and differentiating factors. Culturing cervical cancer cells under this condition can mimic the microenvironment of cervical carcinoma to a large extent [35]. After 14 days of culture, the cells were treated with 300 nM of AZD1152 continuously for an additional 7 days. From the tissue sections (Figure 4), C33A, HeLa, and CaSki were able to differentiate under DMSO treatment, even though the thickness of CaSki cells was relatively thinner than that of C33A and HeLa cells. Treatment with 300 nM of AZD1152 reduced the thickness of all three cell lines. We observed approximately 40–50% of C33A, 10–20% of HeLa, and 20–30% of CaSki cells attached to the epidermal equivalent. It is worth noting that the majority of the HeLa cells treated with 300 nM of AZD1152 appeared to dislodge from the epidermal equivalent. Our results reveal that 300 nM of AZD1152 is sufficient to inhibit differentiation and survival of both HPV-null and -positive cancer cells. 

(ii)HPV-athymic nude mice

As depicted in Figure 5A, we injected HeLa or immortalized primary baby rat kidney (BRK) cells into the dorsal flank of 4–5 weeks old female athymic nude mice. The BRK cells were included as a negative control. This cell line is HPV-null and is able to maintain tumor formation in athymic nude mice, as described in our previous study [36]. After 2 weeks, when a palpable tumor was achieved, the mice were fed with 50 mg/kg of AZD1152 via oral route once every 2 days, for a total of 6 courses within 12 days. The body weight and tumor formation were monitored closely along the course of the experiment (Appendix A). The littles were sacrificed, and tumors were collected and weighed. We observed that the tumor volume of nude mice injected with HeLa cells was reduced dramatically when treated with AZD1152 (34.7% ± 20.1%, *p* < 0.05) compared with DMSO-treated mice (Figure 5B). While tumor volume of mice injected with BRK control cells also reduced significantly upon AZD1152 (47.9% ± 10.0%, *p* < 0.001) treatment (Figure 5C). Taken together, both the results from 3D organotypic culture and animal study consistently show that treatment using AZD1152 inhibits tumor formation of cancer cells, independent of HPV-status of the cells. 

## 4. Discussion

Both E6 and E7 viral oncoproteins are small in molecular mass. However, they possess a robust ability to interact with key cellular proteins and instigate their degradation in a proteasome-dependent manner: E6 degrades p53 and PDZ proteins (hDlg, hScribble, and MAGIs); E7 degrades pRB and its related pocket proteins. These are the most well-known targets of E6 and E7, but the list of their cellular targets is expanding. In this report, we identify for the first time that E6 proteins from cancer-causing HPV types (HPV16 and HPV18) interact with a cell cycle modulator, Aurora kinase B (AurB). We employed a systematic strategy to investigate the oncogenic association between E6 and AurB. Firstly, we investigated the direct interaction between E6 and AurB using biochemical approaches. Secondly, we provided molecular evidence which delineated the consequence of AurB-E6 association to the functions of AurB and their effect in orchestrating oncogenic events of HPV-positive cancer cells. To provide molecular and biochemical evidence on the association between AurB and E6, we utilized transient overexpression using immortalized cancer cell lines with high transfection efficiency. These approaches allow us to study the cellular functional consequences when AurB and E6 were expressed at a 1:1 ratio. Similarly, we were able to study the levels of pAurB and AurB upon E6 depletion at ≥50%. In addition, we focused on delineating the association between E6-encoded by HPV16 and AurB, as HPV16 contributes to the majority of the cancer cases. Thirdly, using biologically relevant preclinical models, we study the specificity and efficacy of commercially available Aurora kinase inhibitors in halting HPV-mediated hallmarks of cancer phenotype. 

Of the Aurora kinase family members, our and previous studies have focused on the association between AurA and HPV [22,27,28], while little is known about the role of AurB in HPV-associated cancers. Although AurB is known to be upregulated and even suggested to be a diagnostic marker in several human cancers, including head-and-neck and cervical cancers [37], the available studies were conducted in disregard of HPV oncoproteins. Our findings provide essential molecular evidence that clearly shows the upregulation of AurB in HPV-immortalized cancer cells correlates directly with the E6 protein level. In this study, we used nocodazole treatment to increase the levels of endogenous AurB. Nocodazole is a microtubule poison and can activate the DNA damage pathway. Upon receiving DNA damage stimuli, due to drug treatment, or upon HPV infection [38], Chk1 is activated. Chk1 then phosphorylates AurB, resulting in its activation and increased expression [39,40] (Figure 6A). We found that the AurB protein corresponds positively with the E6 protein level in HPV-positive cancer cells. During HPV-induced cancer progression, the increased expression of E6 could lead to increased AurB stability and its activity. Of note, the level of AurB is unlikely to be affected by p53, owing to the fact that AurB acts upstream of p53, where AurB phosphorylates p53, leading to the degradation of p53 protein [41,42]. Previous studies have shown that phosphorylation of E6 at E6-PBM upon DNA damage response induction resulted in the inactivation of p53 transcriptional activity [43]. At the same time, the upregulation of AurB mediated by E6 in HPV-containing cervical cancer cells may potentially lead to further inactivation of p53. 

Through the binding assays, we identified that 10 amino acid residues (140 to 150) located immediately upstream of the E6-PBM mediate a previously unidentified oncogenic function of E6. This region is important to form a complex with AurB and AurA [28]. The binding of E6 to AurB may change the conformation of AurB and reduce its ATP binding affinity to AurB (Figure 6B). Even though E6 mutants showed a reduced binding ability to AurB, the weak association between AurB and E6 might be sufficient to perturb the kinase activity of AurB. However, in HPV-positive cells, the association between E6 and AurB increased phosphorylation of AurB, particularly at T232 of AurB. As AurA and AurB share almost 70% homology, they play different roles in cell cycles. AurA plays a central role during interphase and mitotic entry, while AurB is involved in the mitotic phase and cytokinesis. From our previous and current studies, despite binding to the same region of E6, an association of E6 with AurA and AurB appears to have different consequences. We observed that the binding of E6 to AurB reduced AurB kinase activity but did not affect AurA [28]. Nonetheless, the AurA-E6 association could regulate G1/S and mitotic phases of the cell cycle through cyclin E and histone H3 [28]. While AurB-E6 association affects human telomerase, reverse transcriptase (hTERT), and Ras/MEK/ERK signaling axis. Therefore, it is tempting to speculate that when AurA and AurB proteins are aberrantly upregulated in cancer cells, E6 could bind to them simultaneously to abrogate cell cycle checkpoints, allowing cells to survive and proliferate.

AurB is involved in regulating the function of hTERT and Ras/MEK/ERK signaling cascade. We showed that AurB increased hTERT expression, and this is in line with previous findings [25]. AurB, on its own, activates specific protein 1 (Sp1) and c-Myc transcription factors, leading to increased expression of hTERT [25], as depicted in Figure 6A. The upregulation of hTERT could potentially be due to activation of other signaling pathways, post-transcriptional or -translational modification mediated by AurB. This is unclear and worth exploring in the future. In this study, we observed that E6-AurB was essential to maintain a high expression of hTERT protein and its telomerase activity (Figure 6B). Previous studies have shown that the Ras/MEK/ERK axis regulates the expression of both hTERT [44] and AurB [45], while HPVE6 activates the Ras/MEK/ERK pathway [46]. Nonetheless, the expression of hTERT represses the Ras/MEK/ERK pathway [34]. Similarly, we also found that, regardless of the presence of E6 or AurB alone, hTERT protein represses the levels of Ras/MEK/ERK proteins (Figure 6A). Strikingly, a different phenomenon was observed when AurB-E6-hTERT was co-expressed in cervical cancer cells. AurB-E6 association could override the hTERT-mediated suppression on Ras/MEK/ERK pathway (Figure 6B). This could be a mechanism of how E6 exploit AurB to perturb this key oncogenic signaling pathway to enhance cell proliferation and survival. Hence, leading to the increased thickness of the suprabasal layer of HPV-bearing tumor during HPV-induced carcinogenesis.

Upon gaining knowledge about the oncogenic role of AurB in mediating E6-induced oncogenesis, we compared the effect of Aurora kinase inhibition on HPV oncoprotein expression, apoptotic activation, cell proliferation, and tumor formation of HPV-containing and HPV-null cervical cancer cell lines. To date, there is no inhibitor that disrupts the association between AurB and E6. Therefore, we treated these cells with commercially available AZD1152 or Barasertib, a potent and specific AurB inhibitor, which is an ATP-competitive inhibitor currently used in clinical trials for solid tumors. The efficacy and selectivity of AZD1152 on HPV-associated cancers have not been clearly elucidated. Here, we provided preclinical evidence, which unleashes that AZD1152 exhibited a greater efficacy and selectivity in activating apoptotic pathways and inhibiting HPV oncoprotein expression in HPV-containing cervical cancer cells compared to HPV-null cancer cells. The inhibitor also suppressed telomerase activity augmented by AurB and/or E6 (Figure 6C). However, the inhibition is insufficient to suppress telomerase activity in HPV-positive cancer cells, in which hTERT, AurB, and E6 proteins are often produced in high abundance. Consistently, treatment with AZD1152 inhibits proliferation and tumor formation of cancer cells. These effects were observed by disregarding HPV status. Most of the anti-tumor therapeutics available to date exert a broad-spectrum killing effect and often pose adverse after-treatment effects to the patients. As HPV-containing cervical cancer cells might be more susceptible to apoptotic induction, a continuous effort to explore a more target-specific therapeutic option for HPV-associated cancers should be warranted. In addition, it is also worth mentioning that the use of primary keratinocytes to recapitulate the association between AurB and E6 and treatment efficacy should be performed. Our data revealed that the AurB-E6 complex is a drug-able target that could offer a new horizon for the generation of next-generation therapies for HPV-associated cervical cancer and potentially for other HPV-associated cancers.

## 5. Conclusions

In summary, our study shows, for the first time, AurB is a genuine and previously unrecognized target of E6 encoded by cancer-causing HPVs. We also provided mechanistic evidence on how E6-mediates cancer progression through AurB by increasing the hTERT protein level and its telomerase activity to induce cell immortalization, proliferation, and survival. More importantly, we revealed that treatment with AZD1152, a potent and specific AurB inhibitor, inhibited tumor formation of HPV-positive cells by activating the apoptotic pathway. Nonetheless, the inhibitory effect exerted by AZD1152 is independent of HPV status in the cancer cells.

## Figures and Tables

**Figure 1 cancers-15-02465-f001:**
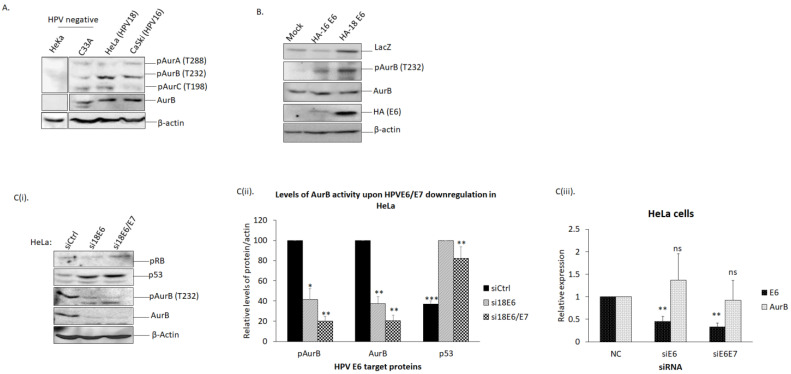
The level of Aurora kinase B correlated positively with the levels of HPV oncoproteins. (**A**) We compared the levels of Aurora kinases in HPV-negative and -positive cells. Total lysate of HPV-negative HeKa (primary human keratinocytes) and C33A (cervical cancer cells), and HPV-positive HeLa (HPV18 cervical cancer cells) and CaSki (HPV16 cervical cancer cells) were collected. The levels of Aurora kinases phosphorylated at threonine (T): Aurora kinase A (AurA) at amino acid 288 [pAurA (T288)], Aurora kinase B (AurB) at amino acid 232 [pAurB (T232)] and Aurora C at amino acid 198 [pAurC (T198)], plus total AurB, were examined via western blotting. Note the increased levels of pAurB (T232) and total Aurora B in HeLa and CaSki. β-actin was included as a loading control. (**B**). The level of phosphor-AurB (pAurB) upon E6 overexpression. HEK293 cells were transfected with HA-tagged 16E6 or HA-tagged 18E6. The cells were also transfected with LacZ as a transfection efficiency control. The total cell lysate was collected and subjected to western blotting. Representative immunoblots showing the levels of phospho-AurB [pAurB (T232)], total AurB, HA (for E6), plus β-actin as a loading control. Note the increased expression level of pAurB upon overexpression of HA-16 and -18 E6. (**C**(i)). The level of pAurB and total AurB were examined upon E6 downregulation. HeLa cells were transfected with siRNA against control (siCtrl), HPV18 E6 (si18E6), or HPV18 E6 and E7 (si18E6/E7). After 72 h, the total lysate was collected, and the levels of pAurB and total AurB were ascertained by western blotting. Also shown is the rescued level of p53, indicating the efficient downregulation of E6. β-actin was included as a loading control. (**C**(ii)). The bar graph shows the quantitation of the levels of pAurB, AurB, and p53 upon HeLa cells transfected with siRNA against control (siCtrl) (black bar), 18E6 (siE6) (green bar) and 18E6/E7 (siE6/E7) (blue bar). Quantitation was performed using ImageJ software, and statistical analysis was performed using Prism. Error bars represent mean ± standard error of the mean (SEM) (n = 3). (ns = not significant, * *p* < 0.05, ** *p* < 0.01, *** *p* < 0.001). (**C**(iii)). The expression of HPV18E6 and AurB genes upon si18E6 or si18E6/E7 in HeLa cells was quantitated by performing a quantitative polymerase chain reaction (qPCR). The level of the respective mRNA was normalized with GAPDH housekeeping control. Error bars represent mean ± standard error of the mean (SEM) (n = 3). (ns = not significant, * *p* < 0.05, ** *p* < 0.01, *** *p* < 0.001) (Western Blots in Appendix A).

**Figure 2 cancers-15-02465-f002:**
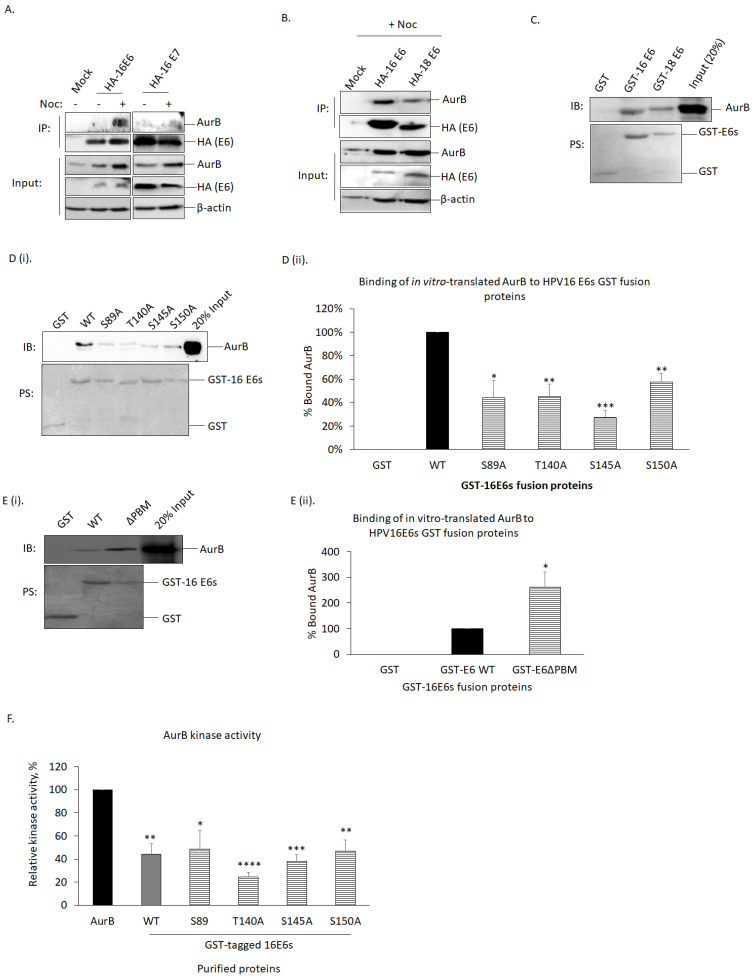
Aurora kinase B bound directly to the C-terminus of E6-encoded by HPV16 and 18 interacted directly in the nucleus or mitotic cells, leading to perturbation of kinase activity of Aurora kinase B. (**A**). Co-immunoprecipitation was performed to determine the binding of Aurora kinase B (AurB) to HPV16E6 or E7. HEK293 cells were either mock-transfected or transiently transfected with HA-tagged HPV16E6 (HA-16E6) or HPV16E7 (HA-16E7). The cells were either treated (+) with nocodazole (Noc) for 18 h to induce increased expression of AurB in the cells or left untreated (−). The cells were then collected and lysed. E6 and E7 were immunoprecipitated using HA-conjugated agarose beads, and the bound AurB was analyzed by western blotting. Note the strong interaction between HPV16E6 with AurB but not with 16E7. β-actin was included as a loading control. (**B**). Co-immunoprecipitation was performed to determine the binding of AurB to HPV16 and 18 E6. By using a similar approach, HEK293 cells were either mock-transfected or transiently transfected with HA-16 E6 or HPV-18 (HA-18 E6). The cells were either treated with nocodazole (+) for 18 h or left untreated (−). The samples were then subjected to western blotting to detect the endogenous AurB precipitated. Note the strong interaction between HPV-E6s with AurB. (**C**). Direct binding assay of AurB and E6. The indicated purified GST fusion proteins were incubated with *in vitro*-translated Flag-tagged AurB (FL-AurB). Following extensive washing, the bound FL-AurB was detected via western blotting. The upper panel shows immunoblot (IB) detected using an AurB-specific antibody; the lower panel shows the Ponceau stain (PS) of the blot. (**D**(i)). Direct binding assay to determine key amino acids of E6 that bind to AurB. By using a similar approach, the indicated purified GST E6 wild type (WT) and mutant fusion proteins were incubated with *in vitro*-translated Flag-tagged AurB (FL-AurB). The upper panel shows immunoblot (IB) detected using an AurB-specific antibody; the lower panel shows the Ponceau stain (PS) of the blot. (**D**(ii)). The bar graph shows the mean levels of bound FL-AurB quantitated using ImageJ software and normalized with the amount of the relative GST fusion proteins in comparison with WT. (**E**(i)). Direct binding assay to determine key amino acids of E6 that bind to AurB. By using a similar approach, the indicated purified GST E6 wild type and E6 with its C-terminus PDZ binding motif depleted (E6ΔPBM) mutant fusion proteins were incubated with *in vitro*-translated Flag-tagged AurB (FL-AurB). The upper panel shows immunoblot (IB) detected using an AurB-specific antibody; the lower panel shows the Ponceau stain (PS) of the blot. (**E**(ii)). The bar graph shows the mean levels of bound FL-AurB quantitated using ImageJ software and normalized with the amount of GST fusion proteins in respective experiments. (**F**). ADP-Glo^TM^ assay was performed to determine the kinase activity of Aurora kinase B (AurB). The indicated purified GST fusion proteins were incubated with purified AurB. The ability of AurB to convert ATP into luminescent ADP was measured. The bar graph shows the AurB activity in percent (%) normalized with the amount of GST fusion proteins in respective experiments from three independent experiments. Error bars represent the mean ± standard error of the mean (SEM) (n = 3). (* *p* < 0.05, ** *p* < 0.01, *** *p* < 0.001, **** *p* < 0.0001). (Western Blots in Appendix A).

**Figure 3 cancers-15-02465-f003:**
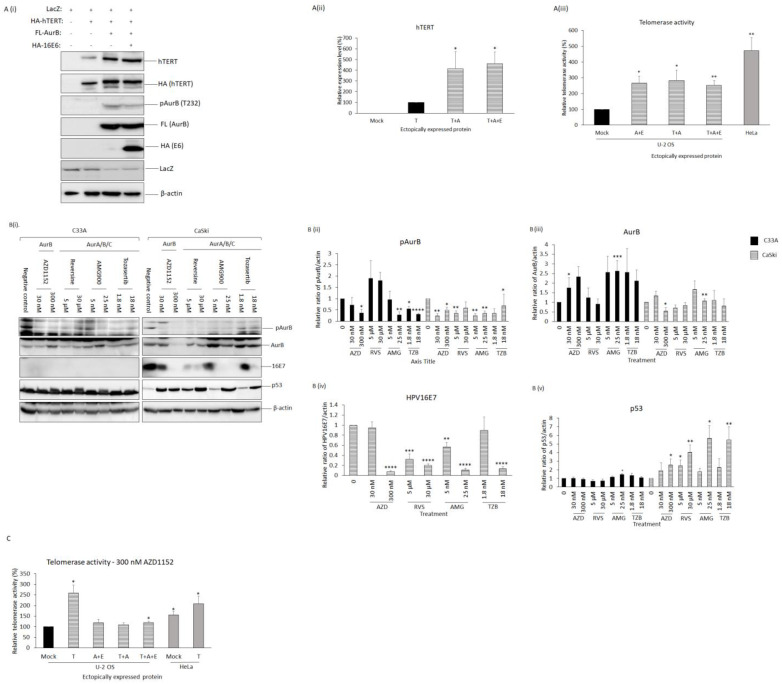
The association of Aurora kinase B and HPV16E6 did not disrupt human telomerase reverse transcription (hTERT) protein expression but induced an increased telomerase activity. (**A**(i)). The levels of hTERT were examined upon co-overexpression of Aurora kinase B (AurB) and HPV16E6 in U-2 OS cells. U-2 OS cells were transiently transfected with either HA-tagged hTERT (HA-hTERT) alone, co-transfected HA-tagged HPV16E6 (HA-16E6), or Flag-tagged AurB (FL-AurB), as indicated. LacZ was included as a transfection efficiency control. The total cell lysate was collected, and the levels of hTERT, phosphor-AurB [pAurB (T232)], AurB, and 16E6 were ascertained via western blotting. Note the increased expression of hTERT upon co-expression with either AurB and/or 16E6. (**A**(ii)). The bar graph shows the mean levels of hTERT protein quantitated using ImageJ software and normalized with loading control in respective experiments. (**A**(iii)). The bar graph shows the quantification of telomerase activity by measuring the signal of all telomerase ladder bands using ImageJ software and by dividing by the intensity of the bands from the internal controls. (**B**(i)). C33A (HPV-null cervical cancer cells) and CaSki (HPV16-positive cervical cancer cells) were treated with a potent Aurora kinase B inhibitor (AZD1152) or pan-Aurora kinase inhibitors (Reversine, AMG900, and Tozasertib) at concentrations indicated for 24 h. Simultaneously, the cells were also treated with DMSO, which served as vehicle control. The total cell lysate was collected, and the expression levels of phosphor-Aurora kinase B (pAurB), total Aurora kinase B (AurB), HPV16E7, and p53 were ascertained via western blotting. Beta-actin (β-actin) was included as a loading control. (**B**) The bar graphs show the mean levels of (ii) pAurB, (iii) total AurB, (iv) HPV16E7, and (v) p53, quantitated using ImageJ software and normalized with loading control in respective experiments. Error bars represent the mean ± standard error of the mean (SEM) (n = 3). (* *p* < 0.05, ** *p* < 0.01, *** *p* < 0.001, **** *p* < 0.0001). (**C**) The bar graph shows the quantification of telomerase activity by measuring the signal of all telomerase ladder bands using ImageJ software and by dividing by the intensity of the bands from the internal controls. (Western Blots in Appendix A).

**Figure 4 cancers-15-02465-f004:**
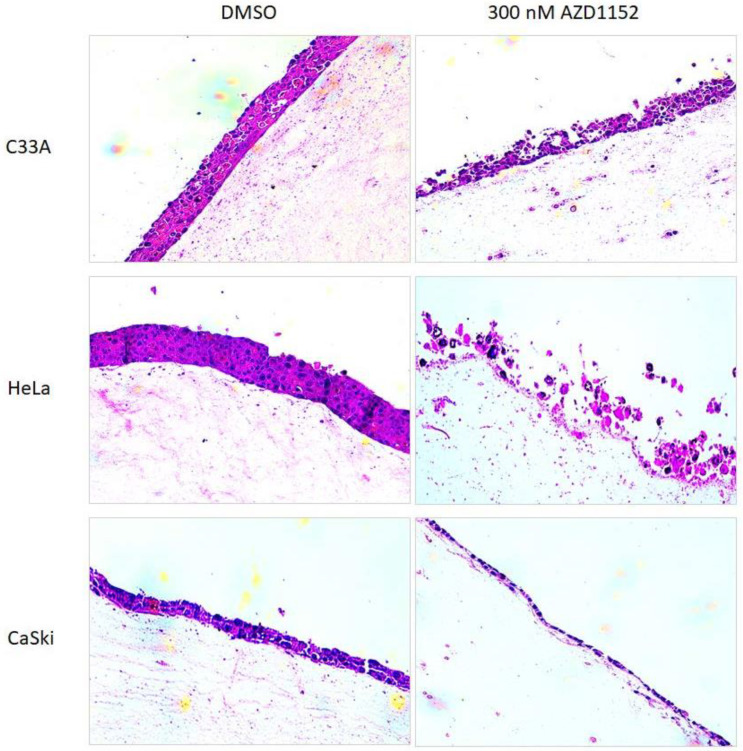
Treatment with AZD1152 inhibited the growth of HPV-positive cancer cells in organotypic culture. Representative images (magnification 200×) show the growth of HPV-positive (HeLa and CaSki) and HPV-null (C33A) treated with 300 nM AZD1152 in a three-dimensional (3D) organotypic culture. As a control, the cells were also treated with DMSO. Note the dramatically reduced thickness of cells upon treatment with AZD1152.

**Figure 5 cancers-15-02465-f005:**
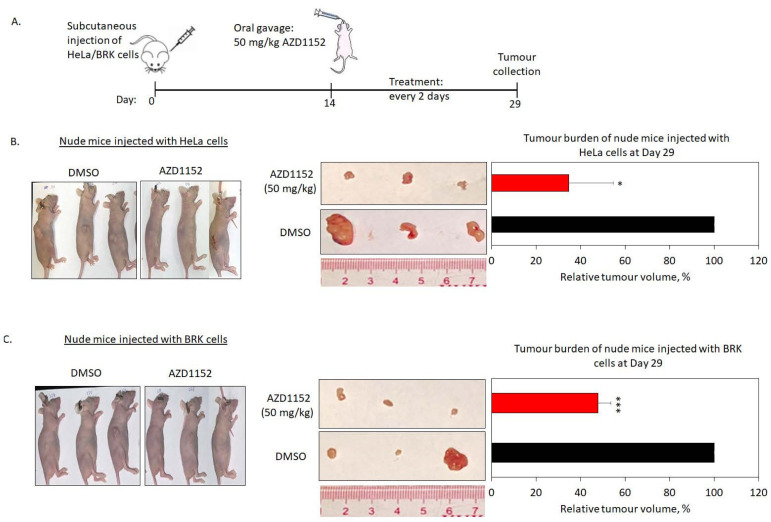
Treatment with AZD1152 inhibited tumor formation of HPV-positive cancer cells in the animal model. (**A**) The schematic diagram shows the experimental setting of the animal study. HPV-positive (HeLa) cells or immortalized baby rat kidney (BRK) cells were injected into at least 12 athymic nude mice subcutaneously. After 14 days, the mice were given 50 mg/kg of AZD1152 via the oral route once every 2 days. Tumors were collected on Day 29. (**B**,**C**). The left panel shows representable images of the nude mice prior to tumor collection. The middle panel shows images of the representable tumor collected. The bar graphs on the right panel show the average percentage of tumor volume (calculated using the formula ab^2^ × 0.5236) in athymic nude mice injected with (**B**) HeLa cells and (**C**) BRK cells treated with AZD1152 (red bars) relative to DMSO-treated (black bars) nude mice collected on Day 29. Data are represented as means ± standard error of the mean (SEM). (* *p* < 0.05, *** *p* < 0.001).

**Figure 6 cancers-15-02465-f006:**
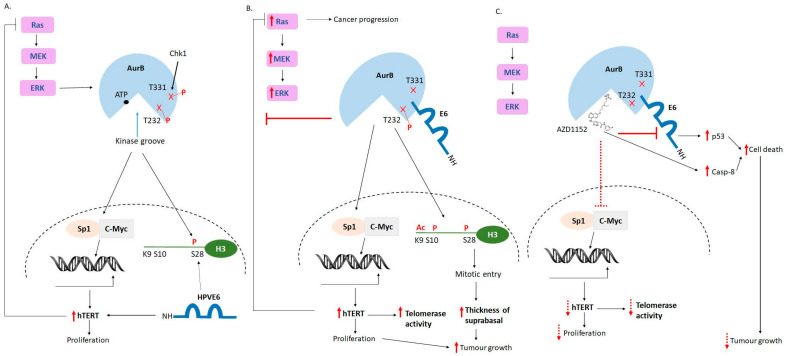
A schematical diagram summarizes the molecular consequences of the association between Aurora kinase B and HPVE6 to cellular events, and treatment of AZD1152 can disrupt the oncogenic event mediated by the AurB-HPVE6 complex. (**A**). Aurora kinase B (AurB) is a kinase with its ATP acceptor site located at amino acid residue 120. Phosphorylation of AurB at amino acid residue threonine (T) 232 and/or T331 increases its kinase activity. AurB is also phosphorylated by Chk1 at T331. HPVE6 and AurB, independently, favor phosphorylation histone H3 at position serine (S) 28. AurB activates transcription factors specific to protein 1 (Sp1) and C-myc, which in turn leads to increased expression (indicated by the solid red arrow) of human telomerase reverse transcriptase (hTERT). E6 also has a role in increasing the hTERT protein level. While increased hTERT protein resulted in increased cell proliferation, hTERT also inhibited the Ras/MEK/ERK axis. (**B**). HPVE6 binds to AurB at the C-terminal end of E6 (residues 140 to 150). This interaction may result in altered protein conformation of Aur. This, in turn, results in the distortion of the ATP binding site within AurB. However, the phospho-status of AurB at T232 remains unaffected. In addition, the association between AurB and HPVE6 did not disrupt the ability of AurB to increase hTERT protein level and its telomerase activity, as well as the post-translational modification of histone H3: acetylation of histone H3 at residue lysine (K) 9 and phosphorylation of histone H3 at residues S10 and S28. In addition, the hindrance effect of hTERT on Ras/MEK/ERK expression is also overridden by the AurB-E6 complex, leading to increased expression of this key oncogenic signaling pathway. As the lesion progresses, the level of E6 protein increases. This will also increase the AurB protein level and activity, leading to cell immortalization, proliferation, and survival, with dysregulated cell division and formation of the eventual carcinoma. (**C**). Treatment with AZD1152, a potent and specific Aurora kinase B inhibitor, suppresses AurB activity and activates the apoptotic pathway. The treatment also can reduce hTERT protein level and its telomerase activity, halting HPV-mediated carcinogenesis, albeit the inhibitory effect occurs in a non-HPV-specific manner (indicated by dashed red arrows).

**Table 1 cancers-15-02465-t001:** Sequence of primer pairs used in real-time PCR for the detection of the respective target genes.

Target Gene	Forward 5′ → 3′	Reverse 5′ → 3′	Reference
E6	aatgtttcaggacccacagg	gttgcttgcagtacacacattc	[31]
AurB	tttctctctaaggatggccc	tctcccttgagccctaagag	[32]
GAPDH	ggtcatccctgagctgaacg	gcctgcttcaccaccttctt	*-*

## Data Availability

Available upon request from the corresponding author.

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
