# Peer review of "Interaction between Human Papillomavirus-Encoded E6 Protein and AurB Induces Cell Immortalization and Proliferation—A Potential Target of Intervention"

_cancers, 2023, doi:10.3390/cancers15092465_

Round 1

Reviewer 1 Report

In this manuscript, Boon et al. propose that interaction between the E6 oncoprotein encoded by the high-risk type HPV and Aurora kinase B (AurB) underlies the virus-induced carcinogenesis. This study is an extension of their work in an earlier published manuscript that claimed HPV-mediated carcinogenesis is mediated via interaction of E6 with Aurora kinase A (AurA). Given this, it would have been interesting if the authors presented a comprehensive hypothesis for the HPV E6 interaction with Aurora kinases in general as they serve as important cell cycle regulators. Nevertheless, the evidence presented in this manuscript strongly suggests that E6 directly interacts with AurB and this complex has the potential for cell immortalization followed by cancer progression. This is supported by the presented evidence for AurB interaction with E6 to increase hTERT levels and telomerase activity as potential mechanisms underlying the carcinogenesis process. However, the manuscript lacks clarity and some of the conclusions presented are based on weaker data, particularly those related to data from in vivo application employing immune deficient preclinical models.

Major concerns:

1.       Data presented in Figure1A is interpreted as “dramatic increase in pAurB in HPV positive cells compared with HPV null cells”: Since total AurB is also increased, the activity of E6 is not just AurB activity, which is important to discuss. Importantly, densitometry is required to quantitate the differences. Furthermore, the primary keratinocyte cell line HaKa used doesn’t’ show any Aurora kinase bands making it difficult to understand the reason for using these cells in this experiment. Additionally, it is desirable to add data form the analysis of additional HPV+ and HPV- cells lines, because data from just two cell lines is insufficient for this conclusion of the role of E6 towards “dramatic increase”.

2.       Figure1B- authors have overexpressed HA-tagged- 16/ 18-E6 in HEK293 cells and showed increase in pAurB levels but, T-AurB levels remain unchanged. But when E6 expression was inhibited with siRNA (as in C-i), levels of both pAurB and total Aurora-B decreased. This seems to contrast the data from 1A data, and it is important to elaborate on this. Also, LacZ was used as transfection control in Figure1B that clearly showed differences in the transfection efficiency in the different cell lines used. Additionally, densitometry is needed to substantiate these claims because of there are subtle changes in the LacZ/HA as well as total AurB (even actin bands seem somewhat at increased density in transfected cells).

3.       In Fig. 2E, the WB are too light to appreciate the differences discussed

4.       Figure2G- relative kinase activity of AurB goes down with WT-GST-E6 samples whereas in Figure 1B they showed HA-16/18-E6Ox resulted in increase in pAurkB activity. This needs to be addressed.

5.       Figure3A- Again LacZ levels are inconsistent in different lanes defeating the purpose of using LacZ as a control for the transfection and expression analyses. It may be important to use β-actin control.

6.       Figure3C(i)- Reversine is used as pan-aurora kinase inhibitor. However, this regent is not an Aurora kinase inhibitor. It’s an MPS-1 or mitotic kinase inhibitor.

7.       It is important to discuss selection of the doses for all the inhibitors in Fig3C(i). No refs or supporting data is included for the selection of the dose range.

8.       Figure S2A- also has problems with LacZ levels not being equal. Total amount of protein levels is also not equal. One of the labels is missing from the panel. Authors claim that in lane4 cells transfected with hTERT+ E6+ AurB show increase in RAS-MEK-ERK pathway. No p-values are mentioned in the bar graph for the last lane. Further, change in values from 1 to 1.3 is insignificant.

9.       FigureS3A- authors have shown caspase 8 blot for the apoptosis pathway as an indicator after treatment with Barasertib. It is not expected that total caspase would change, but only the cleaved caspase should/would – so it is important to clarify whether it is the labeling error.  Nevertheless, it is unclear why caspase 8, a up-stream molecule in the apoptotic pathway, rather than the commonly used down-stream proteins like cleaved caspase 3 is used?  Additionally, it would be useful and necessary to complement this data with apoptosis assays like annexin V staining etc.

10.   For in vivo studies the authors have used BRK cells lines as HPV- cell lines. In their entire manuscript they have changed so many cell lines with respect to HPV- cell lines. It would have made more sense if for HPV- cell lines they have selected HPV- cervical or head and neck cell lines. For examples they could have taken HN31 or FADU or PCI15b as HPV- cell lines for their in-vivo study. Also, the number of mice (n=3) in the study is too small for proper calculations of significance.

11.   Data presented in Fig. 4 using raft cultures shows that AurB inhibition equally effected cells expressing HPV E6 or not, which seems to deviate from the focus of the manuscript connecting AurB and HPV E6. Same concern for data from in vivo studies presented in Fig. 5, where irrespective of HPV E6 expression, AurB inhibition influences tumor growth in the immune deficient model. Furthermore, data from the use of immune deficient model will be difficult to extrapolate for human application. There are good HPV tumor models, well established in the literature that could prove useful to the theme of the manuscript connecting AurB and HPV E6

12.   It is unclear how the specific sites of E6 interaction with AurA (from their earlier publication) and AurB (from current manuscript) are identical (e.g. S89, S145, T140).  It is important to clarify if this true and if so, discuss the importance of E6 interacting with AurA vs AurB or both for carcinogenesis and potential targeting for therapeutic approaches

Reviewer 2 Report

1,The article focuse on the association between E6 encoded by HPV 16 and 18,and AurB. The results provide mechanistic evidence on how E6 promotes cancer progression through AurB,and suggest that Aur B-E6 complex could be therapeutic target for HPV infected cancer.

2,Some figures of the WB in manuscript are not clear enough which should be improve,eg:fig 1 A,C;fig. 2 A.

3,As for the nude mice experiment,tumor histology is important to explain the antitumor mechanism of this inhibitor,which is not included in the manuscript. It is necessary to do histology study.

Reviewer 3 Report

“Human papillomavirus-encoded E6 protein exploits AurB for carcinogenesis – a potential target of intervention” by Boon et al. is a clear and well-written manuscript packed with a lot of data. I especially enjoy the statistical details that was also placed in. However, upon detail examination of the actual data/results, the experimental setup is not well designed, and some key information and controls are still missing.

There seems to be a large reliance on overexpression and western blots. While these are good biochemistry techniques, they can be biased and inaccurate. There is quite a bit of noise in the blotting data. The association between E6 and AurB seems to be there but is not very convincing with just the CoIP data (with nocodazole spike as well). CoIP is a fairly low-resolution technique with potential complex-interactions also picked up, so a higher resolution technique like proximity ligation assay may be better.

My biggest concern is the telomerase activity readout (a.k.a. TRAP assay). There is no mention of the input that is used in the TRAP assay. Typically, this is the number of cells used as input in the TRAP reaction. In the telomerase field, we typically use a range of input (for example, 25, 250, 2500 cells) to ensure that the signal is within the dynamic range. In this paper, the author appears to present the argument that telomerase activity inhibition by AZD1152 treatment appears is through TERT downregulation. However, it Is important confirmed that the treatment itself does not inhibit TRAP signal. The authors also present a one-track explanation for hTERT upregulation through SP1 and c-myc which was not even explored here in this study. There are several paths for TERT regulation (epigenetically, post translational modifications, etc.)!

Some results could be shifted to supplementary section. Figure 2 and 3 are massive figures that could be split apart. For example, Figure 2F are just images without any quantification. Unless there are quantifications, these are lower-level evidence and thus should be moved to supplementary section. Figure 4 also has the same issue and could be moved to supplementary section unless they are quantified.

There are also a couple of comparison in which the control had a very big outlier that was included (Figure S2 and 5). I would be wary of making conclusions with a replicate that is this off from the rest of the data.

While I think the manuscript is written quite clearly and nice to read, the actual experiment and data interpretation needs lots of work. Please refer to the specific points below. I think the evidence for telomerase connection is weak, some of the other results are suggestive of some sort of connection to carcinogenesis. I believe that this manuscript will need to be majorly revised with potential updates to some of the data and overall messaging before it can be considered publish.

Specific points:

Section 3.3 (i) – I am rather surprised that, after transfection with HA-E6 without nocodazole treatment, there isn’t an apparent increase in Co-IP signal for AurB. The input suggest that the AurB levels are still quite high without any treatment. Is nocodazole a requirement for this interaction?

Figure 2 E(i-ii) – It is not very apparent from the immunoblot picture that it is three times higher for AurB. It looks pretty similar to me whereas the GST band actually looked quite different. Please confirm if the calculation is correct in the bar graph. This could change the conclusion of PBM dependency.

Figure 2F – Is the middle two rows just different field of view for 16E6 overexpression? One of the arrows doesn’t seem to show colocalization.

Section 3.4 – The immunocytochemistry staining here is not very convincing. E6 and E7 overexpression seems to produce high signals all over the cells (both cytoplasm and nucleus) so some colocalization is bound to happen. The colocalization in the telophase cells is not very apparent as it may have become just one foci. I would avoid stating the cell cycle phase with just looking at morphology unless if there are also cell cycle markers used.  For E7, it looked like one .of the two cells had colocalization as well. Qualification of colonization using imaging analysis tools like CellProfiler, ImageJ or FIJI would be more reliable. These evidence are all pretty weak, so I would suggest that they be moved into supplementary section or enhanced with qualifications.

Section 3.6 – Isn’t both the hTERT and 16E6 HA-tagged? Is anti-HA antibody used in these western blots? You would be picking up different targets then which could be separate by size but it is still not ideal.

Section 3.6/ Figure 3a – Where is the source of the AurB-mediated hTERT increase from? Exogenous (vector) or endogenous (in the genome)? Does AurB (or A+E) by itself increase hTERT? There has been some evidence in the literature but this should be repeated in this model system as well.

Section 3.7/ Figure 3b – How many cells were used in the TRAP assay? It is very hard to quantify TRAP activity without a concentration sweep of the cells (eg. 5000, 500, 50; in fold changes of 10x, lower fold changes can be used to detect small changes). These results are very inconclusive without controlling the input. Why is the TRAP signal higher with just AurB+E6 without any hTERT added? Given U2OS is a TERT-negative cell line, where is the TERT coming from? This is suggesting change in epigenetic control of TERT. Since HELA is a TERT-positive cell line, I am not sure why it would need hTERT overexpression.

Line 452 – AurB+E6 increase hTERT protein? Where is this data? We cannot be sure whether it is exogenous or endogenous. It is possible that the hTERT overexpression itself is affected.

Section 3.7/Figure S2 – the Ras/MEK/ERK levels of triple co-expression doesn’t look similar at all to mock, it just has very high error bars (driven by outliers). I would not be able to conclude from this result.

Line 477-479 – How are these doses determined? Are these doses commonly used for these compounds?

Section 3.8/Figure 3c – It is interesting that the compounds have such a huge different on different cell lines in regards to pAurB and AurB levels. Is that due to the HPV-positivity in CaSki? It is also strange that the higher doses of some compounds showed higher levels of pAurB (which is opposite of the expected downward trend). 16E7 expression looks pretty nice, but was 16E6 expression also examined?

Section 3.8/Figure 3d – Are these all AZD1152 treated cells? Is there any vehicle control in this experiment? Same thing for the TRAP assay, the quantification is not reliable without some sort of cell concentration sweep. Another way to conduct this experiment is to sweep concentration of AZD1152 with one concentration of cells.  

Section 3.9/Figure 4 – Are there ways to quantify the thickness of these layers? Some image processing tools like ImageJ might be able to quantify these. I suggest at least putting a label with the measurements in the specific images.

Section 3.9/Figure 5 – The differences seem to be largely driven by one large tumor (a potential outlier) in the DMSO group.

Round 2

Reviewer 1 Report

See attached file

Author Response

Please kindly find our response to reviewer's comment in the attachment. 

Reviewer 3 Report

The revised submission of the “Human papillomavirus-encoded E6 protein exploits AurB for carcinogenesis – a potential target of intervention” by Boon et al. has address most of my concerns from the first version. There are still a couple of things I would like addressed:

-Supplementary file: I am unable to see the new version. Is this new version uploaded?

-Figure 4: Please add scale bars to these histo images.

Overall, it is good that the authors chose to soften some of the conclusions based on the results. For a stronger and a more convincing discussion, I suggest that some of the answers to the review questions could also be incorporated as they provide important contexts as well (e.g. inhibitor specificity, developmental effects on tumor response, etc.). 

Author Response

We appreciate the reviewer for the time spent reviewing our revised manuscript and giving us valuable comments. We uploaded a new version of the supplementary material. We also included scale bars for the immunofluorescence images shown in the revised Figure S2.